# Cardiac Remodeling in the Absence of Cardiac Contractile Dysfunction Is Sufficient to Promote Cancer Progression

**DOI:** 10.3390/cells11071108

**Published:** 2022-03-25

**Authors:** Lama Awwad, Tomer Goldenberg, Irina Langier-Goncalves, Ami Aronheim

**Affiliations:** Department of Cell Biology and Cancer Science, The Ruth and Bruce Rappaport Faculty of Medicine, Technion-Israel Institute of Technology, P.O. Box 9649, Haifa 31096, Israel; lamaaw@campus.technion.ac.il (L.A.); tomergo@campus.technion.ac.il (T.G.); irinalan@campus.technion.ac.il (I.L.-G.)

**Keywords:** cardiac remodeling, cancer progression, secreted factors

## Abstract

Cardiovascular diseases and cancer are the leading cause of death worldwide. The two diseases share high co-prevalence and affect each other’s outcomes. Recent studies suggest that heart failure promotes cancer progression, although the question of whether cardiac remodeling in the absence of cardiac contractile dysfunction promotes cancer progression remains unanswered. Here, we aimed to examine whether mild cardiac remodeling can promote tumor growth. We used low-phenylephrine (PE)-dose-infused in mice, together with breast cancer cells (polyoma middle T, PyMT), implanted in the mammary fat pad. Although cardiac remodeling, hypertrophy and fibrosis gene hallmarks were identified, echocardiography indicated no apparent loss of cardiac function. Nevertheless, in PE-infused mouse models, PyMT-cell-derived tumors grew larger and displayed increased cell proliferation. Consistently, serum derived from PE-infused mice resulted in increased cancer cell proliferation in vitro. ELISA and gene expression analysis identified periostin, fibronectin and CTGF as cardiac- and tumor-secreted factors that are highly abundant in PE-infused mice serum as compared with non-infused mice. Collectively, a low dose of PE infusion without the deterioration of cardiac function is sufficient to promote cancer progression. Hence, early detection and treatment of hypertension in healthy and cancer patients would be beneficial for improved outcomes.

## 1. Introduction

Hypertension is a major risk factor for cardiovascular disease morbidity and mortality [1]. Long-term hypertensive heart conditions often occur due to multiple cardiovascular risk factors, including elevated cholesterol levels, reduced high-density lipoproteins, diabetes, obesity and left ventricular hypertrophy [2]. Hypertension does not have warning signs or symptoms and develops over time, and is thus treated only when cardiovascular complications arise. Epidemiological data suggest that the prevalence of hypertension has doubled in the past four decades, apparently because of population aging [3]. Therefore, the elderly population is found to be at higher risk of developing both hypertension and cancer [4]. In addition, it is well established that anti-cancer treatments have adverse cardiac effects, known as cardiotoxicity. As a case in point, hypertension is being recognized as the most frequently recorded comorbidity in patients with cancer [5]. Recent studies indicate that cancer and cardiovascular diseases (CVDs) are connected at various levels [6,7]. These two conditions (CVD and cancer) share similar risk factors, including smoking, unhealthy diet and obesity, physical inactivity, diabetes mellitus, hypertension and alcohol abuse [8]. These factors, alone or in combination, are the main triggers for the development of cancer and are the risk factors for developing CVD [9].

Recent studies have highlighted the crosstalk between CVD and cancer. Interestingly, patients with heart failure (HF) after myocardial infraction (MI) have an increased risk of developing cancer [10].

Consistently, several mouse models of heart failure have been shown to promote cancer progression and metastasis. These studies demonstrated the connection between heart failure/cardiac contractile dysfunction and cancer [11,12,13,14]. Here, we sought to study whether cardiac remodeling in the absence of cardiac contractile dysfunction is sufficient to promote cancer progression. To induce chronic hypertension, mice were infused with low doses of PE (10 mg/Kg/day). We observed that PE-infused mice developed larger tumors compared with control mice. This indicates that hypertension-dependent cardiac remodeling was sufficient to promote cancer progression in the PE-infused mouse models. Collectively understanding the crosstalk between hypertension and cancer is crucial in the field of cardio-oncology in order to provide new strategies to better cope with CVD and cancer and improve patients’ outcomes.

## 2. Materials and Methods

All experimental protocols were approved by the Institutional Committee for Animal Care and Use at the Technion, Israel Institute of Technology, Faculty of Medicine, Haifa, Israel. Approval number IL-144-10-19. All study procedures are complied with the guidelines from of the NIH Guide for the Care and Use of Laboratory Animals.

### 2.1. Animals

Age-matched (8 weeks old) C57Bl/6 female mice were used. Mice were bred and raised at the Pre-Clinical Research Authority at the Ruth and Bruce Rappaport Faculty of Medicine. Experiments were performed following Institutional Animal Care and Use Committee approval and according to the Israeli welfare act, which abides by the National Research Council guidelines, and according to the Guide for the Care and Use of Laboratory Animals of the National Institute of Health. Procedures were carried out under isoflurane anesthesia.

### 2.2. Cell Culture

The PyMT murine breast carcinoma cell line was derived from primary tumor-bearing transgenic mice expressing PyMT under the control of the murine mammary tumor virus promoter [15]. Cells were tested and found to be free of mycoplasma and viral contamination. Cells were cultured in DMEM containing 10% FBS, 1% streptomycin and penicillin, 1% L-glutamine and 1% sodium pyruvate at 37 °C in humidified atmosphere containing 5% CO_2_.

### 2.3. Micro-Osmotic Pumps

Alzet micro-osmotic pumps (#1002, Alzet Cupertino, CA, USA) were filled with phenylephrine (Sigma #SLBN9020V Saint Louis, MO, USA) (10 mg/kg/day, 0.06% acetic acid in saline). Mice were anesthetized with sodium pentobarbital and were subcutaneously implanted with pumps. The procedure was performed under sterile conditions. At the humane endpoint or the end of the experiment, mice were euthanized using isoflurane. Hearts were removed and ventricles were divided into three pieces and used for protein extraction, RNA purification and tissue fixation in 4% formaldehyde overnight, respectively.

### 2.4. Cancer Cell Implantation

PyMT cells were orthotopically injected into the back left side mammary fat pad (10^5^ cells per mouse). Tumor size was measured using a caliper, and tumor volume was calculated with the formula: width^2^ × length × 0.5. The humane endpoint was defined as when the maximal tumor size reached 1500 mm^3^, according to the Institutional Animal Care and Use Committee.

### 2.5. Cell Proliferation in Vitro

PyMT cells were seeded in medium containing 10% FBS at a concentration of 5 × 10^4^ Cells/mL for 6 h. After the cells were attached to the plate, the medium was replaced with serum-free medium and incubated overnight. Next, the medium was replaced with serum-free medium (negative control) and 10% FBS serum in the absence or presence of 10 μg/mL PE, mouse blood serum derived from either control or PE-infused mice. PyMT cells were cultured for additional 48 h. CellTiter-Glo Luminescent Cell Viability Assay was used to measure cell viability according to the manufacturer’s instructions. Luciferase activity was measured with a TD 20/20 luminometer (Turner Designs, Sunnyvale, CA, USA).

### 2.6. Echocardiography

Mice were anesthetized with 1% isoflurane and kept on a 37 °C-heated plate throughout the procedure. Echocardiography was performed with a Vevo2100 micro-ultrasound imaging system (VisualSonics, Fujifilm Toronto, ON, Canada) equipped with 13 to 38 MHz (MS 400) and 22 to 55 MHz (MS550D) linear array transducers. Cardiac size, shape and function were analyzed using conventional 2-dimensional imaging and M-mode recordings. Maximal left ventricular end-diastolic (LVDd) and end-systolic (LVDs) dimensions were measured in short-axis M-mode images. Fractional shortening (FS) was calculated with the following formula: FS% = ((LVDd − LVDs)/LVIDd) × 100. All values were based on the average of at least 3 measurements for each mouse.

### 2.7. RNA Extraction

RNA was extracted from hearts using an Aurum total RNA fatty or fibrous tissue kit (No. 732-6830, Bio-Rad, Hercules, CA, USA) according to the manufacturer’s instructions. Next, cDNA was synthesized from 1000 ng purified mRNA with the iScript cDNA Synthesis Kit (No. 170–8891, Bio-Rad).

### 2.8. qRT-PCR

Real-time polymerase chain reaction analysis was performed with a Rotor-Gene 6000 (Bosch Institute, Sydney, Australia) with the iTaq Universal SYBR green Supermix (BioRad, Hercules, CA, USA). Serial dilutions of a standard sample were included for each gene to generate a standard curve. Values were normalized to either GAPDH or Hsp90 expression levels. The oligonucleotide primer sequence is found in Appendix A.

### 2.9. Blood Serum

Blood was obtained from the facial vein using a 4 μm sterile Goldenrod Animal Lancet (MEDIpoint, Inc., Mineola, NY, USA). Blood was collected and allowed to clot at room temperature for 2 h, followed by 15 min of centrifugation at 2000× *g*. Serum was immediately aliquoted and stored at −20 °C for future use.

### 2.10. Immunohistochemistry

Tumors were fixed in 4% formaldehyde overnight, embedded in optimal cutting temperature (O.C.T) compound, and serially sectioned at 10 μm intervals. Frozen tumor sections were stained for Ki67 (Abcam, ab16667, USA) and counterstained with DAPI. Images were acquired with 3DHistech Pannoramic 250 Flash III (3DHISTECH Ltd., Budapest, Hungary). Each section was fully scanned; for each analysis, 5 fields were randomly chosen and blindly and automatically analyzed with ImageJ software. For every dot plot in the image analysis, each dot represents the mean of the values taken from 5 fields, derived from a single mouse.

### 2.11. Fibrosis Staining

Heart tissue was fixed in 4% formaldehyde overnight, embedded in paraffin, serially sectioned at 10 μm intervals, and then mounted on slides. Masson’s trichrome staining was performed according to the standard protocol. Images were acquired using 3DHistech Panoramic 250 Flash III (3DHISTECH Ltd., Budapest, Hungary). Each section was fully scanned.

### 2.12. ELISA Test

The quantification of periostin, fibronectin and connective tissue growth factor (CTGF) in the serum was performed using the Mouse Periostin/OSF-2 Quantikine ELISA Kit (R&D systems Inc., Minneapolis, MN, USA), Mouse Fibronectin ELISA Kit (E-EL-M0506, Elabscience, and Mouse CTGF-connective tissue growth factor ELISA Kit (E-EL-M0340, Elabscience, Houston, TX, USA), according to the manufacturer’s instructions.

### 2.13. Statistics

Data are presented as mean ± SEM. All mice were included in each statistical analysis unless they were euthanized for humane reasons before the experimental endpoint. Experimental groups were blinded to the experimentalists during data collection. Animals were selected for each group in a randomized fashion. All in vitro experiments were performed in at least four biological repeats and two technical repeats. The statistical significance of the tumor volume was determined via 2-way repeated-measures ANOVA, followed by the Bonferroni posttest. Comparison between several means was performed using 1-way ANOVA, followed by the Tukey posttest. Comparison between two means was performed via Student’s *t*-test, using GraphPad Prism 7 software (La Jolla, CA, USA). Values of *p* < 0.05 were accepted as statistically significant.

## 3. Results

First, we sought to follow cancer progression in a hypertensive mouse model. To induce hypertension, mice were subcutaneously implanted with osmotic mini-pumps filled with low doses of PE (10/mg/kg/day) for four weeks. Subsequently, mice were orthotopically injected with PyMT cancer cells (Figure 1A). Tumor growth was monitored over time until the experiment reached a humane endpoint. We observed increased tumor growth and weight in PE-infused mice as compared with control-tumor-bearing mice only (Figure 1B,C).

The hearts derived from PE-infused mice displayed increased ventricle weight to body weight (VW/BW) ratios compared with those of control mice (Figure 2A). The increase in heart size was accompanied by elevated levels of mRNA corresponding to hypertrophic hallmark gene markers (ANP, BNP) (Figure 2B). The increase in heart size indicates cardiac hypertrophy, which is one of the indications for chronic hypertension. Next, we examined cardiac fibrosis by staining heart sections with Masson trichrome. Heart sections derived from PE-infused mice were stained significantly more strongly compared with heart sections derived from control mice (Figure 2C). Consistently, PE-infused mice showed increased mRNA levels of fibrosis hallmark gene markers (COL1α, TGFβ, ACTA2) (Figure 2D). In addition, the mRNA of inflammatory cytokines IL-1 β and IL6, as well as macrophage cell surface marker F4-80, were found to be elevated in tumor-bearing PE-infused mice (Figure 2E). Importantly, echocardiography performed prior to mouse sacrifice resulted in no apparent cardiac contractile dysfunction, as exemplified by fractional shortening (Figure 2F). In addition, no ventricle wall thickening was observed (Appendix A). Thus, a low dose of PE infusion resulted in cardiac hypertrophy and fibrosis but no cardiac dysfunction. Nevertheless, cardiac remodeling was sufficient to promote cancer growth.

To examine whether the increase in tumor size and weight was due to increased cell proliferation, we stained tumor sections with the cell proliferation marker anti-Ki67. Tumor sections derived from PE-infused mice stained significantly more highly for Ki67, as compared with tumor sections derived from control mice (Figure 3A,B).

To test the role of the putative secreted factor/s responsible for the increase in tumor cell proliferation, serum was obtained from PE-infused and control mice. Mouse serum was used to supplement the serum-free culture medium of growing PyMT cells (10% mouse serum). To exclude a direct effect of PE on PyMT cancer cells; PE was added to FBS-treated cells at 10 μg/mL. The addition of PE displayed no significant effect on PyMT cancer cell proliferation (Figure 3C). Furthermore, PyMT cancer cells supplemented with serum obtained from PE-infused mice displayed higher cell proliferation as compared with serum derived from control mice and fetal bovine serum (Figure 3C).

Collectively, a tumor promotion phenotype was observed in mice infused with low dose of PE. Serum derived from PE-infused mice enhanced cancer cell proliferation in vitro compared with serum derived from control mice.

Previous studies describing the connection between heart failure and cancer suggested the involvement of multiple secreted factors. To study the possible role of the secreted factors in tumor promotion, we used ELISA for the detection of periostin (POSTN), fibronectin (FN) and connective tissue growth factor (CTGF). All these factors were found to be elevated in the serum of tumor-bearing PE-infused mice as compared with control mice (Figure 4A).

Next, we sought to identify the tissue responsible for the production of these factors. We used qRT-PCR with the appropriate primers to identify the expression level of these factors in the mRNA derived from the heart and the tumor. This analysis identified fibronectin and periostin to be elevated in the hearts of PE-infused tumor-bearing mice (Figure 4B). In contrast, the expression levels of connective tissue growth factor (CTGF) were similar in the hearts of PE-infused mice as compared with control mice (Figure 4B). Interestingly, CTGF mRNA levels were higher in the tumors of PE-infused mice (Figure 4C). We also tested the transcription of ceruloplasmin (CP), serpina3, serpin1 and PON1, which were previously identified as putative secreted factors involved in mediating cardiac–tumor crosstalk. Nevertheless, the expression of these factors was similar in both experimental groups (Appendix A).

A schematic summary of the findings of this manuscript is depicted in Figure 5. A cancer promotion phenotype was observed after four weeks PE infusion led to a chronic hypertension phenotype. Intimate crosstalk occurs between the remodeled heart and the tumor, resulting in the secretion of unique factors to the circulation, which are responsible for the increase in tumor progression. Early diagnosis and treatment of hypertension is recommended to obtain better patient outcomes.

## 4. Discussion

Hypertension and cancer have similar risk factors. Hypertension is considered a silent killer and may occur early in life. This condition may be left undiagnosed and untreated as long as no heart failure phenotype is observed. Once diagnosed at advanced stages, patients may experience additional elderly diseases such as cancer. Here we used a mouse model leading to a mild cardiac phenotype after a four-week low-dose PE infusion. This model displayed both cardiac hypertrophic and fibrosis phenotypes, whereas no apparent signs of contractile dysfunction were detected. However, the mild cardiac remodeling observed was sufficient to promote tumor progression as compared with un-treated tumor-bearing mice, independently of cardiac dysfunction. A tumor promotion phenotype was previously described using several mouse models for heart failure, such as myocardial infraction, transverse aortic constriction and ATF3-transgenic mice. The failing-heart–tumor crosstalk was suggested to be mediated via factors secreted to the blood [11,12,14]. We tested three potential tumor-promoting drivers in the serum derived from PE-infused mice and identified periostin, fibronectin and CTGF proteins to have elevated levels in the serum. Subsequently, we found that these factors were expressed at higher levels in the hearts of PE-infused tumor-bearing mice. In contrast, although CTGF was found at high levels in the serum of tumor-bearing PE-infused mice, no increase in the transcription of CTGF was observed in the heart. Surprisingly, CTGF transcription levels were significantly elevated in the tumor and this probably affects tumor growth in an autocrine-dependent manner. These three factors were previously shown to be able to enhance cancer cell proliferation in vitro and in vivo [16,17,18]. Interestingly, using either an ATF3-transgenic hypertrophic model or transverse aortic constriction model, CTGF was highly expressed in the failing heart [12,14]. In contrast, in PE-infused mice that displayed preserved cardiac function, the CTGF expression was elevated solely in the tumor but not in the heart. Nevertheless, the elevated levels of these factors in the circulation are expected to affect both the heart and the tumor. CTGF inhibition by neutralizing antibodies resulted in enhanced left ventricle remodeling repair following myocardial infraction [17] Likewise, FN and periostin expression in the failing heart contributes to cardiac fibrosis and the deterioration of cardiac function [19,20]. On the other hand, increased levels of CTGF, FN and periostin in the circulation are consistent with the increased cancer progression with higher proliferation and metastasis characteristics [21,22,23]. Although the presence of increased levels of these factors may explain the increase in cancer cell proliferation, we cannot exclude the existence of additional secreted factors involved in mediating the cancer progression phenotype. Further experiments are required in order to identify additional factors using proteomics and genomics methodologies. In addition, the increase in inflammatory gene hallmarks in the heart suggests that innate immune cells such as neutrophils and macrophages may play an additional regulatory level in the tumor promotion phenotype [24].

Multiple chemotherapeutic drugs result in hypertension [25]. Increased blood pressure serves as a marker for some chemotherapy treatments [5]. Our results suggest that early treatment, directed towards lowering the blood pressure to normal levels, may be beneficial to prevent tumor progression.

## 5. Conclusions

Hypertension, either as a primary disease or secondary to cancer treatments, may cause the deteriorate of a cancer state. Therefore, early diagnosis and treatment of hypertension is recommended, and would lead to beneficial outcomes for cancer patients.

## Figures and Tables

**Figure 1 cells-11-01108-f001:**
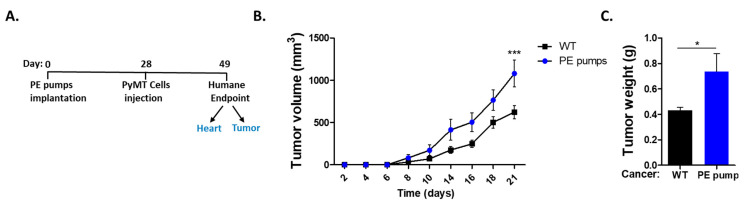
Phenylephrine infusion promotes tumor growth. (**A**) Schematic experimental timeline. Polyoma middle T (PyMT) cells were injected four weeks after PE-infusion (*n* = 4) and compared to control mice (*n* = 4). (**B**) Mice received orthotopic implants in the mammary fat pad with PyMT cells (10^5^ cells per mouse). Tumor volume was monitored over time with the following formula: width^2^ × length × 0.5. (**C**) Tumor weight at the endpoint. Data are presented as mean ± SEM and analyzed via Student’s *t*-test. * *p* < 0.05. *** *p* < 0.001.

**Figure 2 cells-11-01108-f002:**
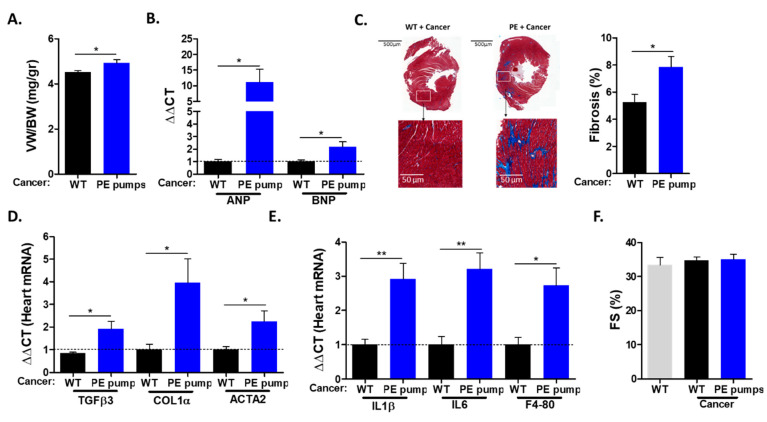
Cardiac remodeling hallmarks following PE infusion. (**A**) Ventricle weight (VW) to body weight (BW) ratio at the endpoint. (**B**) The mRNA levels of hypertrophic markers ANP and BNP were determined via qPCR, normalized with GAPDH. Data are presented as mean ± SEM relative expression compared to control hearts, determined as 1. (**C**) Representative images and quantification of heart sections (*n* = 4, at least 5 fields per mouse) stained using Masson’s trichrome. Scale bar: 500 and 50 µm. (**D**,**E**) Heart mRNA levels of fibrosis markers TGFβ3, collagen 1α and Acta2 (**D**) and inflammation markers IL-1β, IL-6 and F4-80 (**E**) were determined via qRT-PCR, normalized with GAPDH. (**F**) Echocardiography was performed one day prior to sacrifice. Fractional shortening (FS) was assessed according to the formula: FS (%) = ((LVDd − LVDs)/LVDd)). Data are presented as mean ± SEM relative expression compared to control hearts, determined as 1. Data were analyzed via multiple Student’s *t*-tests. * *p* <0.05. ** *p* <0.01.

**Figure 3 cells-11-01108-f003:**
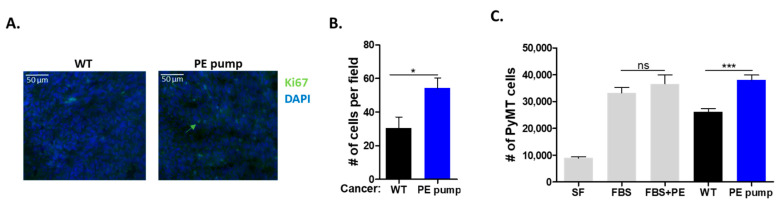
PE infusion increases cancer cell proliferation. (**A**) Representative image of PyMT tumor sections of control and PE-infused mice stained with anti-Ki67 (proliferating cells; green) and DAPI (nuclei; blue). Scale bar: 50 µm. (**B**) Quantification of the number of proliferating cells in tumor section field (*n* = 4, at least 5 fields per each mouse). (**C**) PyMT cells were cultured for 48 h in the absence or presence of 10% FBS either with or without PE at 10 μg/mL, and mouse blood serum drawn from either control or PE-infused mice. Proliferation was measured using a Luminescent Cell Viability Assay using serum from at least three mice per group. Data are presented as mean ± SEM and were analyzed using either Student’s *t*-test (**B**) or one-way repeated-measures ANOVA, followed by Tukey posttests (**C**). * *p* < 0.05. *** *p* < 0.001.

**Figure 4 cells-11-01108-f004:**
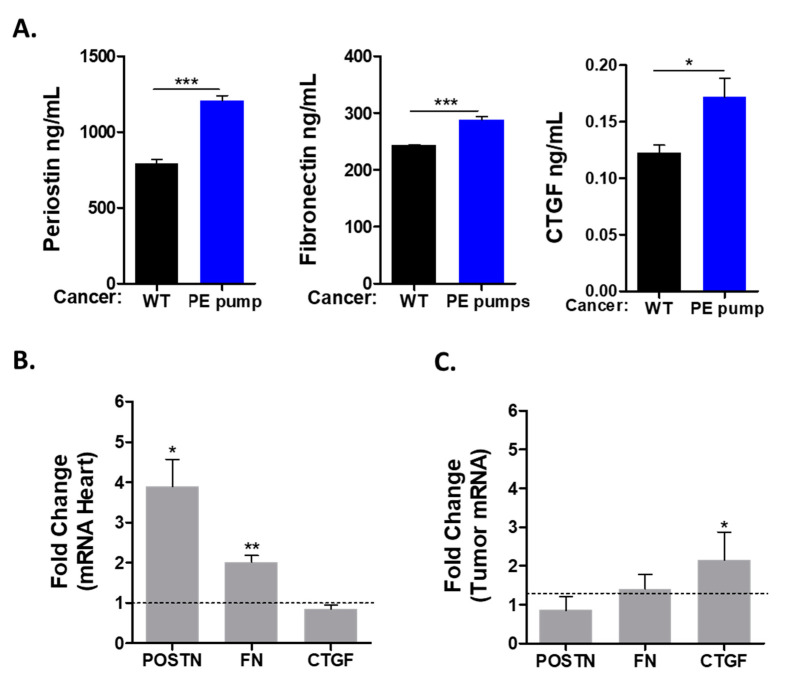
The tumor promotion phenotype is mediated by secreted factors, expressed in either the heart or the tumor. (**A**) Serum levels of periostin, fibronectin and CTGF were determined via ELISA. (**B**,**C**) mRNA levels from either the heart (**B**) or tumor (**C**) of the indicated genes were measured by means of qRT-PCR, normalized with GAPDH (heart) or Hsp90 (tumor) house-keeping genes. Data are presented as relative expression compared with control, determined as 1. Data are presented as mean ± SEM and were analyzed via multiple Student’s *t*-tests. * *p* < 0.05. ** *p* < 0.01. *** *p* < 0.001.

**Figure 5 cells-11-01108-f005:**
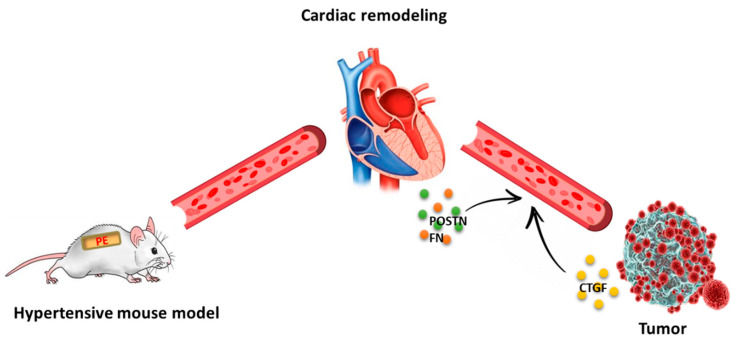
Cardiac remodeling in the absence of cardiac contractile dysfunction is sufficient for tumor promotion. The PE-infusion model. Cancer cells were implanted four weeks after the implantation of PE-infusing pumps. Cardiac remodeling occurred prior to tumor implantation with no apparent cardiac contractile dysfunction, although tumor progression was readily observed. Cardiac (orange and green dots) and tumor-secreted factors (yellow dots) are responsible for tumor growth in a paracrine and autocrine fashion.

## Data Availability

Data related to this manuscript is contained within the article or Appendix A.

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
