# Peer review of "Cardiac Remodeling in the Absence of Cardiac Contractile Dysfunction Is Sufficient to Promote Cancer Progression"

_cells, 2022, doi:10.3390/cells11071108_

Round 1

Reviewer 1 Report

Authors investigate to identify the association between hypertension and cancer. It is interesting that tumor grows factors are secreted under high blood pressure and preserved cardiac function. However, the reviewer has some concerns. Please consider and answer with the reviewer’s questions and comments.

  1. The reviewer afraid that PE has some effect directly on tumor cells. Please consider adding the phenylephrine treated FBS group in Figure 3C to exclude this concern.
  2. The upregulation of periostin and fibronectin in the heart was unclear. Did they contribute to heart remodeling or cancer developing? Please add author’s consideration in discussion.
  3. Please show the age and sex of mice in 2.1. Animals.
  4. Please transfer the primer list to the supplement data.
  5. Please show the scale bar in Figure 2C and 3A and add the scale in the legend.
  6. Please add “C” in Figure 4C.
  7. Do authors have blood pressure data? If yes, please add the supplement data.
  8. Please show LVDd/s in supplement data. In addition, please add IVSd and LVPWd measurements if authors have them.

Author Response

We wish to thank the reviewers for their valuable comments. Please find below point-to-point response to the reviewer comments. 

Reviewer #1 

  1. The reviewer afraid that PE has some effect directly on tumor cells. Please consider adding the phenylephrine treated FBS group in Figure 3C to exclude this concern.

Response: This is a very well taken point. To exclude this possibility, we added PE at 10mg/mL for 48 h on PyMT cancer cells and this bar is included in Figure 3C.

  1. The upregulation of periostin and fibronectin in the heart was unclear. Did they contribute to heart remodeling or cancer developing? Please add author’s consideration in discussion.

Response: Periostin and Fibronectin elevation in the heart affects the heart locally as well as it’s elevation in the circulation results in increase in cancer cell proliferation. This point is clarified in the discussion section of the revised manuscript.

  1. Please show the age and sex of mice in 2.1. Animals.

Response: The age and sex information was added.

  1. Please transfer the primer list to the supplement data.

Response: Primer list was transferred to the supplemental Table 1.

  1. Please show the scale bar in Figure 2C and 3A and add the scale in the legend.

Response: scale bars were added as requested.

  1. Please add “C” in Figure 4C.

Response: The missing label was added.

  1. Do authors have blood pressure data? If yes, please add the supplement data.

Response: The blood pressure measurement in mice is cumbersome and we do not feel confident enough to include them in the manuscript.

  1. Please show LVDd/s in supplement data. In addition, please add IVSd and LVPWd measurements if authors have them.

Response: Echocardiography data is shown in supplemental Table 2.

Reviewer 2 Report

The manuscript entitled ‘Cardiac remodeling in the absence of cardiac contractile dys- 2

function is sufficient to promote cancer progression’ investigates if mild cardiac remodeling can promote tumor growth. The authors used PE infusion to induce cardiac remodeling. Further, cancer cell implantation molecular and histological analysis was performed to test the hypothesis. However, a few aspects that require further attention:

  1. The provided data suggest that PE infusion induces some cardiac remodeling, and the cancer cells proliferate faster. Still, no data shows that increased cancer cell proliferation is due to cardiac remodeling. Therefore, it would be better to use transgenic models for hypertension or hypertrophy without chemical induction.
  2. Though the biochemical markers for hypertension and fibrosis are up, the author should also provide the systolic function analysis (echo) to show the establishment of hypertension.
  3. Likewise, the cardiac cell size measurement should be performed to demonstrate cardiac hypertrophy.
  4. Supplementary figure 1 can go with fig 2; however, a baseline control should be added.

Author Response

We wish to thank the reviewers for their valuable comments. Please find below point-to-point response to the reviewer comments. 

Reviewer #2

  1. The provided data suggest that PE infusion induces some cardiac remodeling, and the cancer cells proliferate faster. Still, no data shows that increased cancer cell proliferation is due to cardiac remodeling. Therefore, it would be better to use transgenic models for hypertension or hypertrophy without chemical induction.

Response: This information is included in our recent publication in Cancer Research. We added this point in the discussion to highlight this work as well.

  1. Though the biochemical markers for hypertension and fibrosis are up, the author should also provide the systolic function analysis (echo) to show the establishment of hypertension.

Response: Echocardiography data is shown in supplemental Table.

  1. Likewise, the cardiac cell size measurement should be performed to demonstrate cardiac hypertrophy.

Response: Cardiomyocytes cell size measurements were not performed.

  1. Supplementary figure 1 can go with fig 2; however, a baseline control should be added.

Response: Supplemental Figure 1 was moved to Figure 2F. A baseline control was included.

Round 2

Reviewer 1 Report

The authors answered sincerely and correctly. The reviewer's concerns were resolved.

Reviewer 2 Report

No further query.